# Pretravel plans and discrepant trip experiences among travelers attending a tertiary care centre family travel medicine clinic

Jacqueline K. Wong [1,2¤a] *, Nancy Nashid[1,2¤b], Lisa G. Pell[3], Ray E. Lam[2], Debra M. Louch[2], Michelle E. Science[1,2], Shaun K. Morris[1,2,3]

**1** Department of Paediatrics, University of Toronto, Toronto, Ontario, Canada, **2** Division of Infectious Diseases, Hospital for Sick Children, Toronto, Ontario, Canada, **3** Centre for Global Child Health, Hospital for Sick Children, Toronto, Ontario, Canada

¤a Current address: Division of Pediatric Infectious Disease, Department of Pediatrics, McMaster Children's Hospital, Hamilton, Ontario, Canada
¤b Current address: Division of Pediatric Infectious Diseases, Department of Pediatrics, Children's Hospital at London Health Sciences Centre, London, Ontario, Canada
* wongj37@mcmaster.ca

**Data Availability Statement:** The SickKids Research Ethics Board approval for this study does not permit posting participant data online. De-identified individual participant data that underlie

## Abstract

### Background

International travel can expose travelers to a number of health risks. Pretravel consultation (PC) helps mitigate risk and prepare travelers for health concerns that might arise. The assessment of risk, mitigation strategies, and relevance of pretravel advice is dependent on how closely travelers adhere to their planned travel itinerary and activities. We determined the proportion of returned travelers whose completed travel experiences differed from their stated travel itineraries, and identified discrepancies that significantly altered the traveler's health risk and would have required alternative counseling during their PC.

### Methods

We conducted a prospective cohort study at the SickKids' Family Travel Clinic between October 2014 and November 2015. Returned travelers who completed a post-travel survey were included. Pretravel consultation assessments and post-trip surveys were compared to identify discrepant trip experiences.

### Results

A total of 389 travelers presented to the clinic for a PC during the study period and 302 (77.6%) were enrolled. Post-travel surveys were received from 119 (39.4%) participants, representing 101 unique itineraries. The median participant age was 36.3 years (IQR 26.6–47.5) and there were 73 female travelers (61%). Most participants (n = 87,73%) were healthy as well as Canadian born (n = 84, 71%). A quarter of travelers were visiting friends and relatives (VFR) (n = 30, 25.2%). The vast majority of returned travelers (n = 109, 92%)

the results reported in this article (text, tables, figures) will be made available to investigators whose secondary data analysis study protocol has been approved by an independent research ethics board. Proposals should be directed to grace. baiano@sickkids.ca; to gain access, data requestors will need to sign a data access agreement.

**Funding:** This work was supported in part by a 2014-2015 International Society of Travel Medicine Research Award to SKM. The funders had no role in study design, data collection and analysis, decision to publish, or preparation of the manuscript. Other funding was provided by SKM's research start-up funds. There was no additional external funding received for this study.

**Competing interests:** The authors have declared that no competing interests exist.

reported discrepant trip experiences involving trip duration, countries visited, accommodations, environmental surroundings and/or activities. Almost two thirds of these individuals (n = 68, 62%) would have required alternative pretravel counseling. We did not identify any demographic or planned trip characteristics that predicted discrepant trip experiences requiring alternative pretravel counseling.

## Conclusions

The majority of travelers reported discrepant trip experiences and the discrepancies often affected health risk. Therefore, clinicians should consider providing broader counselling during the PC as discrepancies from planned travel are common.

## Introduction

International travel can expose travelers to a number of health risks, which vary depending on the trip itinerary and individual traveler factors. While visiting a different region of the world or a specific geographic region within a country, travelers may be exposed to different mosquitoes and vector-borne diseases. Certain activities such as water activities or animal excursions may subject travelers to bodily harm that they would not encounter at home. Furthermore, travelers may engage in high-risk behaviours including unprotected sex and illicit drug use while abroad and expose themselves to additional infectious risks from sexually transmitted infections [1]. Any traveler, including people returning to their home country to visit friends and relatives (VFR) may underestimate risks associated with travel [2]. As a group, VFRs experience higher incidences of travel-related infectious diseases which is partially a result of broader risk exposures such as staying in homes and living the local lifestyle and also considering themselves immune to certain travel-related infectious diseases [2, 3]. Lastly, underlying medical conditions may predispose an individual to more severe outcomes after certain infections that are acquired abroad including malaria and salmonella.

The Pretravel Consultation (PC) offers a dedicated time to prepare travelers for health concerns that might arise during their trips. In addition to obtaining the traveler's medical history, the PC assessment should cover details of the upcoming trip including duration of travel, reason for travel, VFR status, countries to be visited, environmental surroundings, accommodations and special activities (e.g. disaster relief, mountain climbing, diving, etc.) [3]. Clinicians working in travel clinics provide personalized pretravel advice to mitigate potential risks, by highlighting the likely exposures, reminding travelers of ubiquitous risks, and prescribing targeted interventions (such as vaccines and prophylactic medications). Their assessment is predicated on the accuracy and quality of the information provided by the traveler during the PC [4, 5]. Providing appropriate pretravel advice for travelers to adhere to is a key element in ensuring that international travelers return home in good health [6].

The relevance of the pretravel advice and preventative measures are inherently dependent on whether travelers adhere to their stated travel plans. To our knowledge, there has been no published data to date describing the frequency of discrepant trip experiences (i.e. differences between planned itineraries and actual experiences). As such, we sought to determine the proportion of returned travelers whose completed travel experiences differed from their planned itineraries and whether the discrepancies would have altered the traveler's health risk in such a way that alternative pretravel counseling was required. In addition, we explored whether specific demographic or trip characteristics may have predicted discrepant trip experiences

requiring alternative counseling. Lastly, we explored if individuals with discrepant trip experiences also endorsed partaking in high-risk behaviours. It is anticipated that this information may help guide clinicians who provide PCs.

## Methods

### Study site

The Hospital for Sick Children (SickKids) is Canada's second-largest freestanding children's hospital and is a tertiary-care center located in downtown Toronto. Toronto is one of the most diverse cities in the world with 46.1% of the city's population being foreign-born [7]. The SickKids Family Travel Clinic was established in 2013 with the goal of providing pretravel care and recommendations to children and their families [8].

### Study design

This discrepant travel experience study was conducted as part of a larger study [9] assessing risk perception and adherence to recommendations from the PC. The study, which was designed as a prospective cohort, was conducted at the SickKids Family Travel Clinic with enrollment over a 57-week period from October 2014 to November 2015. Per routine clinic practice, travelers were asked to complete a pretravel questionnaire (S1 File) prior to their initial PC, to document information about their demographics, health history, and upcoming travel plans. Individuals were approached for study consent and enrollment when the pretravel questionnaire was distributed. Those who provided consent to participate in the discrepant travel experience portion of the study were contacted by e-mail and invited to complete an online post-travel survey about their actual travel experience and engagement in any high-risk behaviours (S2 File). The online survey request was sent out 1 week post-travel via an online questionnaire administered through REDCap. If the survey was not completed, an initial email reminder was sent 48hr after the original message, and then a phone call reminder was provided 48 hours thereafter. If after an additional 48 hours the survey was not completed, the participants were called and a trained research assistant administered the survey over the phone. The study protocol was approved by the SickKids Research Ethics Board, (REB no. 1000045900).

### Study participants

This study included individuals who attended the SickKids Family Travel Clinic for their initial PC and who completed the online post-travel survey. We excluded hospital employees and those who planned to travel for longer than 1 year. Consent for participation was obtained over the phone or in-person at the PC.

### Data collection

All patients who completed both the pretravel questionnaire and post-travel survey were included in the analysis. The PC questionnaires were reviewed for demographic and medical history information, as well as details of the planned trip itinerary including the duration, reason for travel, countries to be visited, environmental settings to be visited, accommodations and planned activities. If a traveler indicated that they were visiting friends and/or relatives in addition to traveling for other purposes, they were categorized as a VFR. Details pertaining to the actual trip experiences were obtained from the post-travel surveys. If participants were traveling together with the same itinerary (i.e. belonging to the same traveling unit, including families and other groups), their completed trip details were analyzed together. Post-travel

survey responses indicating high-risk behaviours were defined as excessive alcohol consumption (exceeding national low-risk alcohol consumption guidelines [10]), recreational drug use, new tattoos or piercings, or new sexual partners during travel.

Discrepant trip experiences were defined as any difference between stated travel plans (i.e. responses from the pretravel questionnaire) and actual trip experiences (i.e. responses from the post-travel survey). Responses from the pretravel questionnaire and post-travel survey were compared for each participant. For each discrepancy, the impact on risk (i.e. higher, completely different, or lower/no change in risk) and the need for alternative pretravel counseling were defined *a priori* (Table 1) and based on consensus discussion between 3 authors (JKW, NN, SKM). Discrepancies needing alternative counseling included those where the change in risk was higher or completely different. If alternative counseling was required, the actual trip experience was reviewed to determine if additional vaccinations, chemoprophylaxis or other empiric medication prescriptions (e.g. for altitude sickness) would have been indicated.

### Statistical analysis

Demographic information and completed travel experiences were summarized using standard descriptive statistics. Discrepant trip experiences including those requiring alternative pretravel counseling were summarized using counts and proportions. We explored whether the following demographic or trip characteristics were associated with the need for alternative pretravel counseling: age, sex, medical comorbidities, country of birth, VFR status, region of travel, traveling alone, traveling with children, reason for travel, duration of travel, having a fixed itinerary, joining an organized tour, endorsing high-risk behaviours. Univariable analyses to determine the relationship between these characteristics and the need for alternative counseling were conducted using the Spearman's correlation coefficient for continuous variables and chi-squared or Fisher's exact test for dichotomous variables. An exploratory multivariable analysis was planned that included variables with a *p*-value of < 0.2 from the univariable analysis and using step-wise backward elimination. Statistical analyses were performed using IBM SPSS Statistics for Macintosh, Version 26.0, and *p*-values less than .05 were considered statistically significant.

## Results

### Traveler characteristics

During the study period, there were 883 visits to the SickKids Family Travel Clinic. Of the 389 individuals who were eligible for participation, 302 (77.6%) were enrolled in the parent study. Pretravel questionnaires were available for 297 (98.3%) travelers. From this cohort, 119 travelers (40%) completed the post-travel survey and were included in this study on discrepant trips (Fig 1).

The median age of participants meeting inclusion criteria was 36.3 years (IQR 26.6–47.5 years), and there were more females than males (n = 73, 61%). Children under the age of 18 accounted for 13% (n = 15) of participants, with 7 being under the age of 5 years. Almost all individuals (n = 112, 94%) indicated they were traveling with others, with other family members being the most common travel companion (n = 69, 58%). The majority of the travelers were born in Canada (n = 84, 71%). VFRs made up a quarter of travelers (n = 30, 25%). (Table 2)

Comparatively, those who completed the post-travel survey were older (mean age 36.7 years vs. 18.4 years, p<0.001), more often female (61% vs. 46%, p < 0.05), and less likely to identify as a VFR (25% vs. 38%, p < 0.05) than those who only completed the baseline

**Table 1. Decision algorithm for categorizing discrepancies in trip characteristics.**

| Variable (Trip Characteristic) | Discrepancy Details | Change in Risk | | |
|---|---|---|---|---|
| | | Higher | Completely Different | Lower or No change |
| Trip Duration | 25% or more increase in duration | X | | |
| | 25% decrease in duration | | | X |
| | Discrepancy results in having needed additional/different counselling and/or intervention(s)? | | Yes | No |
| Countries Visited (world bank income tiers) | New country (lower income tier than any pretravel country) | X | | |
| | New country (same income tier as any pretravel country OR new income tier but not lower than any pretravel countries OR new income tier and higher than any pretravel countries) | | | X* |
| • high-income | | | | |
| • upper-middle | Removal of country | | | X |
| • lower-middle | Discrepancy results in having needed additional/different counselling and/or intervention(s)? | | Yes | No |
| • low-income | | | | |
| Accommodations | Adding: locals/family/friends, camping or safari | X | | |
| • hotel (any star) | Adding: hostel | | X | |
| • hostel | Adding: hotel, rented house/apt, and/or cruise | | | X |
| • locals/family/friends | Changing from hotel, rented house/apt or cruise ⟺ anything else | X | | |
| • rented house/apt | | | | |
| • camping or safari | Changing between hostel ⟺ locals/family/friends ⟺ camping or safari | | X | |
| • cruise | | | | |
| • other | Changing between hotel ⟺ rented house/apt ⟺ cruise | | | X |
| | Changing between: camping ⟺ safari | | | X |
| | Changing from anything ⟺ hotel or cruise alone | | | X |
| | Changing or adding "Other" (but not described) | . | . | . |
| | Removing anything | | | X |
| | Discrepancy results in having needed additional/different counselling and/or intervention(s)? | | Yes | No |
| Destination | Addition: high altitude, rural/remote, jungle/forest | X | | |
| • urban | Addition: beach or urban | | | X |
| • rural or remote | Changing between: high altitude ⟺ rural/remote ⟺ jungle/forest | | X | |
| • high altitude | | | | |
| • beach | Removing anything | | | X |
| • jungle or forest | Discrepancy results in having needed additional/different counselling and/or intervention(s)? | | Yes | No |
| Activities | Adding: climbing/trekking, water (snorkel, swimming or scuba), raft or boat, animals, cave, school/hospital/orphanage, motorcycle/scooter | X | | |
| • biking | | | | |
| • hiking | | | | |
| • climb/trek | Changing between: climbing/trekking ⟺ water (snorkel, swimming or scuba) ⟺ raft or boat ⟺ animals ⟺ cave ⟺ school/hospital/orphanage ⟺ motorcycle/scooter | | X | |
| • water (snorkel, swimming or scuba considered similar) | | | | |
| • raft or boat | Adding: biking, hiking, public transport | | | X |
| • animals | | | | |
| • cave | Removing anything | | | X |
| • public transport | Discrepancy results in having needed additional/different counselling and/or intervention(s)? | | Yes | No |
| • school/hospital/ orphanage | | | | |
| • motorcycle/scooter | | | | |
| OVERALL | Any discrepancies that results in having needed additional/different counselling and/or intervention(s)? | Yes (if any above = yes) | | No (if all of above are No) |

* When adding a new country within the same income tier, the new country needed to belong to the same geographical region as other countries in the itinerary to avoid introducing different infectious diseases risks

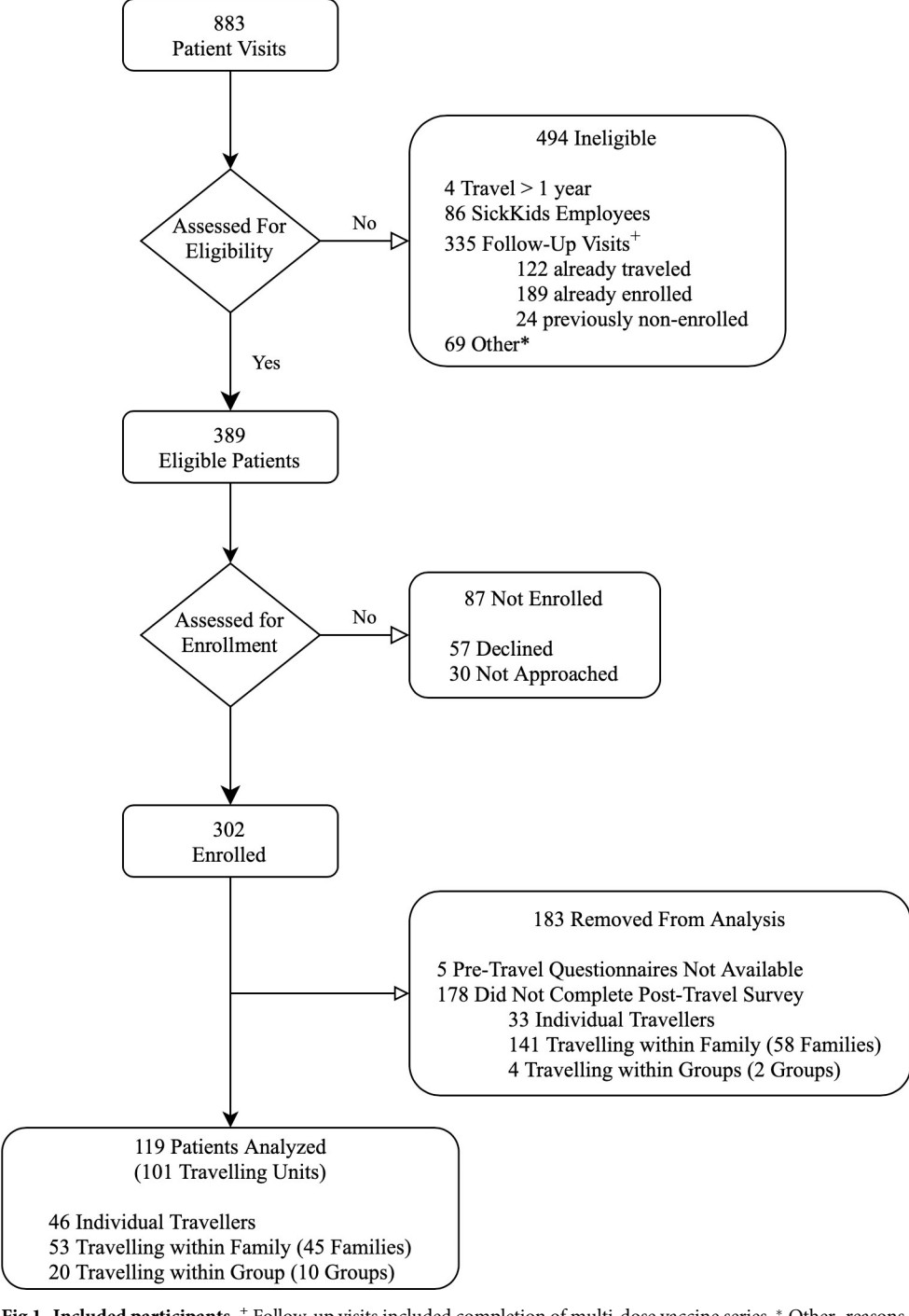

**Fig 1. Included participants.** [+] Follow-up visits included completion of multi-dose vaccine series. [*] Other–reasons for ineligibility not documented.

pretravel questionnaire. Two thirds of participants who did not complete the post-travel survey were children under the age of 18. Half of these pediatric participants (n = 65, 54%) belonged to a traveling unit (group or family) where at least one other member had provided a post-travel questionnaire.

**Table 2. Demographics and planned travel characteristics of travelers at the SickKids Travel Clinic for October 2014 to November 2015.**

| | Completed Post-Travel Survey (N = 119) n (%) | Baseline Travel Survey Only (N = 178) n (%) |
|---|---|---|
| Age * | | |
| < 5 | 7 (6) | 41 (23) |
| 5–10 | 3 (3) | 43 (24) |
| 11–17 | 5 (4) | 35 (20) |
| 18–35 | 43 (36) | 19 (11) |
| 36–55 | 47 (39) | 37 (21) |
| > 55 | 14 (12) | 3 (2) |
| Mean (SD) | 36.7 (16.6) | 18.4 (16.9) |
| Median (IQR) | 36.3 (26.6–47.5) | 11.7 (5.4–33.0) |
| Gender ** | | |
| Male | 46 (39) | 93 (52) |
| Female | 73 (61) | 85 (48) |
| Country of Birth | | |
| Canada | 86 (72) | 141 (79) |
| Outside of Canada | | |
| North America | 3 (3) | 2 (1) |
| Caribbean | 2 (2) | 0 (0) |
| South & Central America | 4 (3) | 0 (0) |
| Europe | 8 (7) | 12 (7) |
| Mediterranean | 1 (1) | 4 (2) |
| Africa | 5 (4) | 7 (4) |
| South Asia | 3 (3) | 6 (3) |
| Southeast Asia | 2 (2) | 1 (1) |
| Pacific | 5 (4) | 5 (3) |
| VFR ** | 30 (25) | 67 (38) |
| Traveling Alone | | |
| Yes | 7 (6) | 4 (2) |
| No Details | 0 | 1 |
| Travel Insurance | | |
| Purchased/Intending to purchase | 99 (86) | 135 (82) |
| Not specified | 4 | 13 |
| Comorbidities | | |
| None | 87 (73) | 141 (79) |
| Mental health | 5 (4) | 6 (3) |
| Chronic Hepatitis | 2 (2) | - |
| Asthma | 5 (4) | 5 (3) |
| Gastrointestinal | - | 2 (1) |
| Dyslipidemia | 4 (3) | 1 (1) |
| Hypertension | 4 (3) | 3 (2) |
| Thyroid Condition | 4 (3) | 1 (1) |
| Musculoskeletal | 4 (3) | 2 (1) |
| Chronic Kidney Disease | - | 1 (1) |
| Diabetes | - | 2 (1) |
| Sickle Cell | - | 1 (1) |
| Immunocompromised | 4 (3) | 6 (3) |

(*Continued*)

**Table 2.** (Continued)

|  | Completed Post-Travel Survey | Baseline Travel Survey Only |
|---|---|---|
|  | (N = 119) | (N = 178) |
|  | n (%) | n (%) |
| Other | 1 (1) | 4 (2) |
| Smoker | 3 (3) | 2 (1) |
| No Details | 3 | 6 |

[*] p < 0.001

[**] p < 0.05

## Actual trip characteristics

The 119 completed surveys represented 101 unique travel itineraries (Table 3). Included among the individual respondents were 30 individuals belonging to 3 families and 10 groups. The median trip duration was 17 days (IQR 11–22) and 15 trips lasted greater than 1 month (n = 14%). The most common reasons for travel were for vacation purposes (n = 58, 57%) and visiting friends and/or relatives (n = 30, 30%). Most accommodations were hotels (n = 81, 80%); however, almost half of the trips included a stay with either locals or friends and/or families for a part of the trip (n = 40, 38%). Forty-four itineraries (44%) included multiple types of accommodations. Two-thirds of the pretravel itineraries indicated that a single country would be visited (n = 68, 67%). The most commonly visited regions were South and Central America (n = 31, 31%), Africa (n = 17, 17%) and the Caribbean (n = 17, 17%). Travelers visited a variety of settings including urban regions (n = 91, 90%), rural and/or remote regions (n = 70, 69%) and beaches (n = 67, 66%). Most individuals participated in more than one type of activity or excursion (n = 85, 71%) that could have exposed them to health risks, whereas 9 (8%) reported partaking in no activities.

## Discrepant trip experiences

Of the 119 travelers who completed post-travel surveys, 109 (92%) reported discrepant experiences with 76 individuals having discrepancies in more than one trip characteristic (Fig 2). Sixty-eight individuals (62%) would have required alternative pretravel counseling due to an increase or different type of risk than anticipated based on the pretravel questionnaire. Changes to planned activities (n = 97) and visited environmental surroundings (n = 66) were the most common discrepancies, and the most common to result in the need for alternative pretravel counseling (54% and 41%, respectively). The majority of these scenarios would not have resulted in different recommendations for pre-travel medication prescriptions. However, three travelers (2.8% of those with discrepant experiences) may have benefited from additional medication recommendations (i.e. malaria prophylaxis while camping or visiting a jungle environment, and altitude sickness preventative medications). Numerous scenarios lacked sufficient detail about actual travel to determine if there would have been a change in pre-travel medication recommendations (n = 21 scenarios, affecting 27.9% of travelers who would have required alternative counseling). In particular, unexpected animal exposure (a new activity exposure) was reported by 17 travelers; however, because questions about the details of the animal exposure were not specifically asked (e.g. a bite from a wild urban dog vs. a visit to an elephant sanctuary), we were unable to determine if pre-travel recommendations would have differed. The regression analysis did not reveal any traveler demographic or trip characteristic that predicted the need for alternative counseling.

**Table 3. Completed trip characteristics.**

| | n (%)[#] |
|---|---|
| Unique Trip Itineraries (N = 101) | |
| Trip Duration (Median Duration in Days, IQR) | 17 (11–22) |
| Reason for travel (could indicate more than one)[^] | |
| Vacation | 58 (57) |
| VFR[$] | 30 (30) |
| Education or Business | 7 (7) |
| Volunteering/Humanitarianism | 4 (4) |
| Cruise | 3 (3) |
| Religious reasons/Pilgrimage | 0 (0) |
| Adoption | 1 (1) |
| Other/Not indicated | 2 (2) |
| Long-stay travel[+] | 15 (15) |
| Organized Tour (Yes or Partial) | 44 (44) |
| Number of Countries Visited | |
| 1 | 68 (67) |
| 2 | 20 (20) |
| 3 | 8 (8) |
| 4 | 4 (4) |
| 5 or more | 5 (5) |
| Regions Visited (Could choose more than one)[$] | |
| Caribbean | 17 (17) |
| South & Central America | 30 (30) |
| Europe | 1 (1) |
| Eastern Mediterranean | 2 (2) |
| Africa | 15 (15) |
| South-East Asia | 10 (10) |
| South Asia | 14 (14) |
| Western Pacific | 12 (12) |
| Accommodations (Could choose more than one) | |
| Hotel | 81 (80) |
| Hostel | 10 (10) |
| Locals/Friends/Family | 40 (40) |
| Rented House/Apartment | 19 (19) |
| Camping or Safari | 12 (12) |
| Cruise Ship | 6 (6) |
| Environmental Surroundings (Could choose more than one) | |
| Urban | 91 (90) |
| Rural/Remote | 70 (69) |
| Beach | 67 (66) |
| Jungle/Forest | 47 (47) |
| High Altitude | 21 (21) |
| Individual Travelers (N = 119) | |
| Activities | |
| Biking | 13 (11) |
| Hiking (Hiking, Climbing) | 59 (50) |
| Water Related (Snorkeling, Swimming, Scuba) | 56 (47) |
| Boating (Boating, Rafting) | 49 (41) |

(*Continued*)

**Table 3.** (Continued)

|  | n (%)# |
| --- | --- |
| Contact with Animals | 43 (36) |
| Caving | 5 (4) |
| Public Transit | 70 (59) |
| Visiting Schools, Hospitals or Orphanages | 27 (23) |
| Motorcycle or Scooter Use | 13 (11) |
| High-Risk Behaviour (N = 104 adults) |  |
| Any alcohol consumption | 83 (80) |
| Alcohol consumption exceeding safe limits* | 13 (16) |
| Not enough information to quantify | 10 (12) |
| Recreational drug use | 3 (3) |
| New tattoos or piercings | 1 (1) |
| New sexual partner | 3 (3) |

# Unless otherwise specified

^ There were 29 VFRs who also indicated other reasons for travel as follows: 26 vacation, 1 vacation & pilgrimage, and 3 for other reasons (not otherwise specified). There were 6 non-VFRs who indicated more than one reason for travel as follows: 3 vacation & cruise, 2 vacation & education or business, 1 education or business & volunteering/ humanitarianism

§ Visiting friends and/or relatives; If a traveler indicated that they were visiting friends and/or relatives in addition to traveling for other purposes, they were defined as a VFR for this study. There were 26 VFR who indicated they were traveling for vacation purposes, 1 VFR who was traveling for vacation and pilgrimage purposes, and 3 VFR who were traveling for other purposes not specified

± Defined as one month or longer

$ Visited countries categorized based on The World Bank Country and Lending Groups [11]

* 10 standard drinks a week for women and 15 drinks a week for men (Canada's Low-Risk Alcohol Drinking Guidelines) [10]

Among the 104 adult respondents, less than one fifth (n = 17, 16%) reported engaging in high-risk behaviours. Three quarters of these individuals reported excessive consumption of alcohol (n = 13, 76%). There were no significant demographic differences between travelers who reported engaging in any high-risk behaviours and those who did not.

## Discussion

To our knowledge, this is the first study to describe discrepant trip experiences among attendees at a family travel medicine clinic. Travelers who completed the post-travel survey were mostly young, healthy adults. A quarter of individuals self-identified as VFRs. The median trip duration was just over 2 weeks, and most often for vacation purposes to a single country. Discrepancies between planned and completed trips were very common and varied across trip characteristics. These differences often altered the health risk to the traveler and would have required alternative pretravel counseling. We did not identify any traveler characteristics that predicted the need for alternative counseling.

International travel has been increasing globally, and in Canada there has been consecutive year-over-year increases since 2012 [12]. VFR travel contributes a substantial amount of tourism worldwide with up to 50% of travel to certain regions by VFR travelers [2, 13]. In the recent definition by the Migration Health Sub-Committee of the International Society of Travel Medicine, 'a VFR traveler is a traveler whose primary purpose is travel to visit friends or relatives, where there is a gradient of epidemiological risk between home and destination'

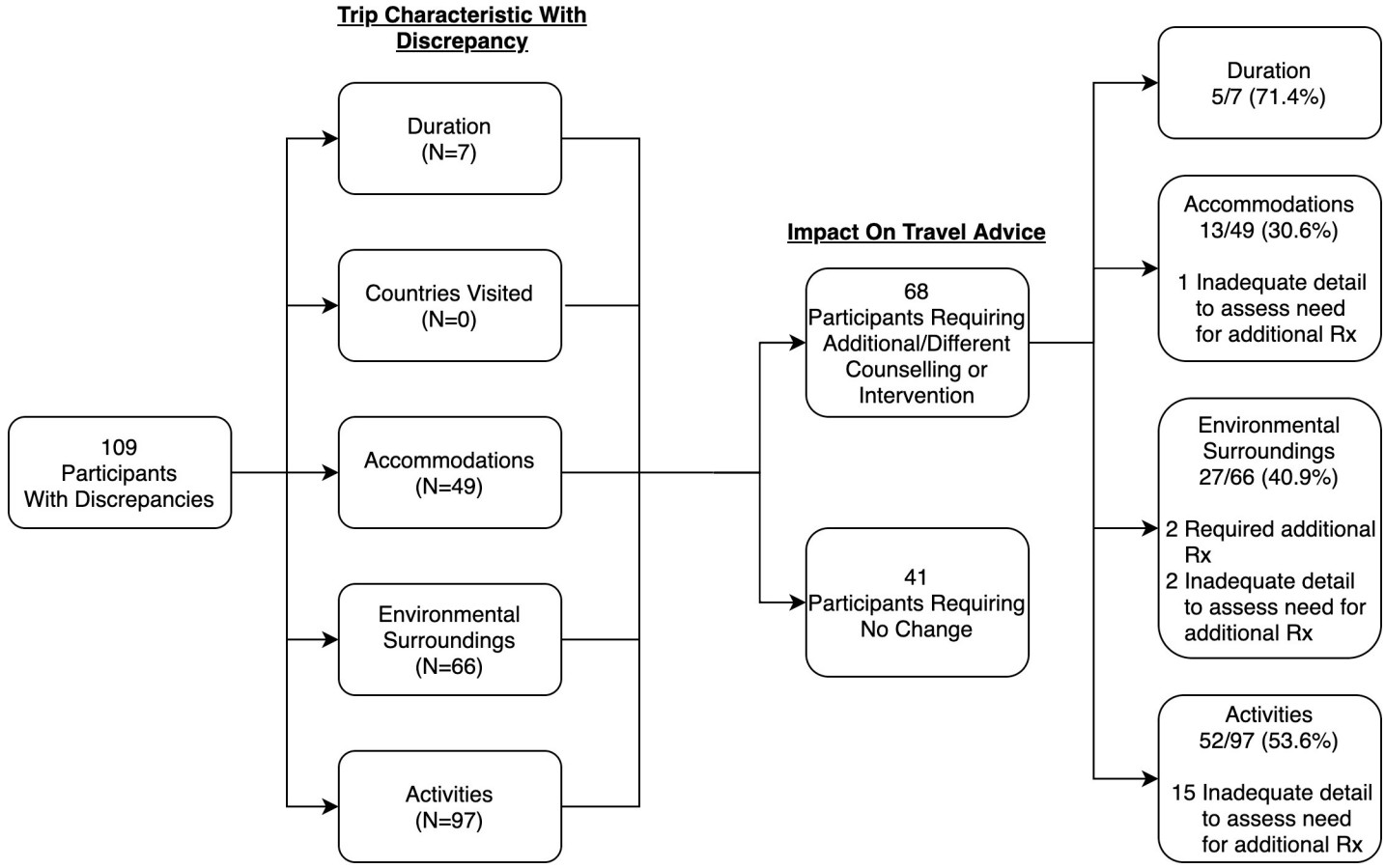

**Fig 2. Travelers reporting discrepant trips (counts/proportions) (N = 119).** * Participants could report discrepancies in >1 trip characteristic. A total of 76 individuals had discrepancies in more than one trip characteristic.

[14]. This group is often under-represented in travel-related studies, yet these travelers are at increased risk of travel-related illness [15]. In our study, VFRs made up the minority of participants who completed the post-travel survey. We found that VFR status did not predict either discrepant trip experiences nor discrepancies that would require alternative PC counseling. However, we were unable to assess trip discrepancies for two thirds of the VFR population visiting the clinic during the study period who did not complete the post-travel survey.

Published data examining travel-related health risks to children is limited [16]. Their reason for travel and associated risks usually depend on the adult with whom they are traveling. During the study period, almost half of all travelers who enrolled in the parent study were children aged less than 18 years (n = 136, 46%). Though post-travel surveys were only completed for 15 children, there were 65 children who travelled as part of a family or group unit where another adult returned their post-travel survey. If we assumed that children travelled and stayed in their travel unit and analyzed these additional responses, a total of 56 children (70%) had trip discrepancies in at least one of trip duration, countries visited, accommodations or destinations visited. Twenty-eight of these discrepancies (50%) would have required alternative pretravel counseling. Traveling with a child did not influence the likelihood of discrepant trip experiences or the need for alternative pretravel counseling.

There were limitations to our study. Firstly, patients who choose to obtain a PC are inherently different from those who do not, perhaps reflecting a difference in risk perception and

behaviours when traveling. In addition, the cohort who responded to the survey may not have been representative of the complete pre-travel population seen by our clinic, as evidenced by the differences in those who were male, children and identified as VFR (Table 2). Though it is unclear if these demographic differences may have correlated with differences in risk perceptions, it was unlikely this introduced a relevant selection bias given we found the vast majority of travelers had trip experiences that were inconsistent with their planned itineraries and these differences often altered the travel-related health risks to the individual. Secondly, there are limited data to objectively adjudicate the change in risk as a result of a modification in travel itinerary, and the decision algorithm was developed by clinicians based on their clinical experiences and rational judgment. For example, even though World Bank income rankings were used to attribute and compare travel-related risks for different countries, these would not capture differences in durations spent in specific countries or the specific regions that were visited within a given country. Furthermore, the responses from the post-travel survey often lacked sufficient detail to determine if the changes impacted travel within certain endemic illness zones in a given country or if additional vaccinations, prophylaxis or other medications would have been indicated based on the nature of the exposure (i.e. animal exposures). Lastly, we were unable to identify any predictors for the need for alternative pretravel counseling. Many travelers did not complete a post-travel questionnaire and the final size of our cohort limited the robustness of the regression analysis.

## Conclusion

We described the travel experiences for a diverse group of travelers who obtained a PC at a family travel medicine clinic located in a busy and multicultural urban setting. The vast majority of travelers reported discrepant trip experiences, with most introducing novel and/or increased risks to their health. Therefore, the advice provided in the original PC may not have been optimal. This study informs practitioners providing pretravel advice to consider broader counselling as discrepancies from planned travel are common.

## Supporting information

**S1 File. SickKids family travel clinic pre-travel health consultation and history form.** (PDF)

**S2 File. Post-travel survey (acute occurrence of risk).** (PDF)

## Acknowledgments

This work was presented in part at the 56th Annual Meeting of the Infectious Diseases Society of America (IDWeek); 2018 October 4; San Francisco, CA; Abstract 456.

## Author Contributions

**Conceptualization:** Shaun K. Morris.

**Data curation:** Jacqueline K. Wong, Nancy Nashid.

**Formal analysis:** Jacqueline K. Wong.

**Funding acquisition:** Shaun K. Morris.

**Methodology:** Lisa G. Pell, Ray E. Lam, Debra M. Louch, Michelle E. Science, Shaun K. Morris.

**Supervision:** Shaun K. Morris.

**Visualization:** Jacqueline K. Wong.

**Writing – original draft:** Jacqueline K. Wong.

**Writing – review & editing:** Jacqueline K. Wong, Nancy Nashid, Lisa G. Pell, Ray E. Lam, Debra M. Louch, Michelle E. Science, Shaun K. Morris.

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
