## [Decision Letter · Decision Letter 0]

28 Jun 2021

PONE-D-20-38921

Pretravel plans and discrepant trip experiences among travelers attending a tertiary care centre family travel medicine clinic

PLOS ONE

Dear Dr. Wong,

Thank you for submitting your manuscript to PLOS ONE. After careful consideration, we feel that it has merit but does not fully meet PLOS ONE’s publication criteria as it currently stands. Therefore, we invite you to submit a revised version of the manuscript that addresses the points raised during the review process.

Please review and consider all points raised by reviewers to strengthen the acceptability of this manuscript for publication consideration.  In addition, please recommend the following changes:

- figure 2 can be incoporated into table 2 or table 3 for descriptive purposes.

- please address in discussion/limitations section the potential bias of the study sample who completed post-travel and the attendant limitations. 

- please address whether the changes would have resulted in different vaccination, chemoprophylaxis, or other empric medication prescriptions (motion sickness, altitude, anti-diarrheals, lepto prophy, etc) and the frequency which this would have needed to occur.   While it is appreciated that the focus was on pre-travel counseling, it's difficult to entangle this part of the travel medicine practice from provision of vaccines and medications.

We look forward to receiving your revised manuscript.

Kind regards,

Mark Simonds Riddle

Academic Editor

PLOS ONE

Journal Requirements:

 [This work was supported in part by a 2014-2015 International Society of Travel Medicine Research Award to SKM. The funders had no role in study design, data collection and analysis, decision to publish, or preparation of the manuscript.]. 

Reviewers' comments:

Reviewer's Responses to Questions

**Comments to the Author**

1. Is the manuscript technically sound, and do the data support the conclusions?

Reviewer #1: Yes

Reviewer #2: Yes

2. Has the statistical analysis been performed appropriately and rigorously? 

Reviewer #1: I Don't Know

Reviewer #2: Yes

3. Have the authors made all data underlying the findings in their manuscript fully available?

Reviewer #1: No

Reviewer #2: Yes

4. Is the manuscript presented in an intelligible fashion and written in standard English?

Reviewer #1: Yes

Reviewer #2: Yes

5. Review Comments to the Author

Reviewer #1: Main comments: The authors have compared pretravel consultation information and and post-trip surveys from 119 participants, finding that 92% reported discrepant trip experiences, two thirds of which would have required alternative pretravel counselling. While there are no major methodological problems, there are a relatively small number of participants, and the impact and potential biases related to losses to follow-up between pre- and post-travel cases need to be better discussed.

Also, the results may be of some interest to readers, but the clinical relevance and details of the changes to the pretravel counselling that would have been warranted based on changes in trip characteristics should be listed so that their importance can be assessed.

Minor issue: Line 102 says that consent was obtained in writing, and then line 115 says that consent was obtained over the phone or in-person. Why the discrepancy?

Reviewer #2: I would recommend the paper for publication.

The paper addresses the subject of travel and pretravel counseling at a very timely moment when more countries are relaxing travel restrictions and individuals are again making International travel plans. The information contained in the sample that was collected during a period of unrestricted travel from October 2014 and November 2015 will be comparable to, and reflective of, the anticipated unrestricted travel environment that is expected in the coming months and years.

The study design is sound and the diversity of the patient population benefits the study. The post-travel follow-up seems adequate and the authors did not lose many patients in follow-up. With a median age of 36.3 years, the study population does skew younger and it would have been advantageous to include an older population, especially people who take cruise vacations.

It is clear that the frequency of changing travel plans necessitates pre-travel counseling, but the study would be improved if there was longer follow-up, especially whether the change in travel plans meant the traveler would have needed a different immunization regime. It would have also been helpful to know if the change in travel plans impacted which endemic illness zone the traveler entered and the effect it had on their required post-travel medications. The time constraint involved with each pre travel visit, will preclude extended counseling required and perhaps the author could discuss this in detail more.

Overall, the study will help identify changing travel patterns among travelers and the health risk associated with it.

6. PLOS authors have the option to publish the peer review history of their article (what does this mean?). If published, this will include your full peer review and any attached files.

Reviewer #1: No

Reviewer #2: No

---

## [Author Response · Author response to Decision Letter 0]

21 Sep 2021

Editor’s Comments: 

Please review and consider all points raised by reviewers to strengthen the acceptability of this manuscript for publication consideration. In addition, please recommend the following changes:

-Thank you for reviewing our manuscript and providing the opportunity to resubmit a revised version. 

Figure 2 can be incorporated into table 2 or table 3 for descriptive purposes.

-We agree that the information in Figure 2 can be concisely incorporated elsewhere. Table 3 (Completed Trip Characteristics) has been updated to reflect this change. 

Please address in discussion/limitations section the potential bias of the study sample who completed post-travel and the attendant limitations. 

-This is an important limitation to provide further elaboration, and additional details have been included in the discussion section (lines 311-315). We agree that those who completed post-travel surveys likely differed from those who did not, as there were differences in the demographic characteristics between these two groups. The current literature assessing the impact of various traveler characteristics on risk perceptions is limited. 

Please address whether the changes would have resulted in different vaccination, chemoprophylaxis, or other empiric medication prescriptions (motion sickness, altitude, anti-diarrheals, lepto prophy, etc.) and the frequency which this would have needed to occur. While it is appreciated that the focus was on pre-travel counseling, it's difficult to entangle this part of the travel medicine practice from provision of vaccines and medications.

-We appreciate the opportunity to provide additional details that may help clinicians who read our manuscript. Regarding over the counter medications (such as those for motions sickness and anti-diarrheals), these would normally have been discussed in the clinical encounter however were not captured in the pre-travel counseling research data. Furthermore, questions in the post-travel survey were not developed to include the details necessary to determine if these additional medication recommendations would have been warranted (e.g. amount of time spent on a boat or the type of boat). Therefore, these medications have not been included in the discussion of specific medication recommendations.

-For the other prescription medications (vaccinations, malaria prophylaxis, altitude-sickness medications) these are now included in the methods section (lines 154-157). The impact on pre-travel medication prescriptions has been detailed in lines 251-263, and Figure 2 has been updated as well. In some scenarios, our assessment was limited by a lack of sufficient detail about actual travel or activities provided in the post-travel surveys, and this limitation has been included in the discussion (lines 322-325). The majority of the time however, the discrepancies would not have resulted in additional medications/vaccinations being prescribed. 

Reviewers’ Comments: 

Reviewer #1: Main comments: The authors have compared pretravel consultation information and post-trip surveys from 119 participants, finding that 92% reported discrepant trip experiences, two thirds of which would have required alternative pretravel counselling. While there are no major methodological problems, there are a relatively small number of participants, and the impact and potential biases related to losses to follow-up between pre- and post-travel cases need to be better discussed.

-We thank this reviewer for their comments. As suggested in the Editor’s comments, the limitations due to cohort size and loss to follow-up has been further elaborated on in the discussion section (lines 311-315).

Also, the results may be of some interest to readers, but the clinical relevance and details of the changes to the pretravel counselling that would have been warranted based on changes in trip characteristics should be listed so that their importance can be assessed.

-Translating our findings into clinically relevant information has been a focus of this revised manuscript. In addition to the impact on pre-travel medication prescriptions suggested by the Editor’s comments, we also elaborated on the frequency of unexpected animal exposure (details in lines 251-263).

Minor issue: Line 102 says that consent was obtained in writing, and then line 115 says that consent was obtained over the phone or in-person. Why the discrepancy?

-We thank the reviewer for identifying this inconsistency. The clarification has been provided in line 115.

Reviewer #2: I would recommend the paper for publication.

The paper addresses the subject of travel and pretravel counseling at a very timely moment when more countries are relaxing travel restrictions and individuals are again making International travel plans. The information contained in the sample that was collected during a period of unrestricted travel from October 2014 and November 2015 will be comparable to, and reflective of, the anticipated unrestricted travel environment that is expected in the coming months and years.

The study design is sound and the diversity of the patient population benefits the study. The post-travel follow-up seems adequate and the authors did not lose many patients in follow-up. With a median age of 36.3 years, the study population does skew younger and it would have been advantageous to include an older population, especially people who take cruise vacations.

It is clear that the frequency of changing travel plans necessitates pre-travel counseling, but the study would be improved if there was longer follow-up, especially whether the change in travel plans meant the traveler would have needed a different immunization regime. It would have also been helpful to know if the change in travel plans impacted which endemic illness zone the traveler entered and the effect it had on their required post-travel medications. The time constraint involved with each pre travel visit, will preclude extended counseling required and perhaps the author could discuss this in detail more.

Overall, the study will help identify changing travel patterns among travelers and the health risk associated with it

-We thank this reviewer for their comments and are appreciative of the suggestion for future study improvements, including longer study follow-up and discussions around post-travel medication counseling. They have identified important opportunities to elaborate on the discrepant travel plans and the need for alternative immunization regimens as similarly mentioned in the Editor’s comments (details now included in lines 251-263). With regards to the clinically relevant limitation of knowing the change in travel plans and impact on endemic illness zones, this has also been included in the discussion section (lines 322-325).

---

## [Editor Report · Decision Letter 1]

19 Dec 2021

Pretravel plans and discrepant trip experiences among travelers attending a tertiary care centre family travel medicine clinic

PONE-D-20-38921R1

Dear Dr. Wong,

We’re pleased to inform you that your manuscript has been judged scientifically suitable for publication and will be formally accepted for publication once it meets all outstanding technical requirements.

Kind regards,

Mark Simonds Riddle

Academic Editor

PLOS ONE
---

## [Editor Report · Acceptance letter]

25 Jan 2022

PONE-D-20-38921R1 

Pretravel plans and discrepant trip experiences among travelers attending a tertiary care centre family travel medicine clinic 

Dear Dr. Wong:

I'm pleased to inform you that your manuscript has been deemed suitable for publication in PLOS ONE. Congratulations! Your manuscript is now with our production department. 

Kind regards, 

on behalf of

Dr. Mark Simonds Riddle 

Academic Editor

PLOS ONE